# Isolated Hypomethylation of *IGF2* Associated with Severe Hypoglycemia Responsive to Growth Hormone Treatment

**DOI:** 10.3390/diagnostics11050749

**Published:** 2021-04-22

**Authors:** Sarah C. Grünert, Uta Matysiak, Franka Hodde, Gunda Ruzaike, Ekkehart Lausch, Anke Schumann, Natascha van der Werf-Grohmann, Ute Spiekerkoetter, Miriam Schmidts

**Affiliations:** Department of General Paediatrics, Adolescent Medicine and Neonatology, Faculty of Medicine, Medical Centre—University of Freiburg, 79106 Freiburg, Germany; uta.matysiak@uniklinik-freiburg.de (U.M.); franka.hodde@uniklinik-freiburg.de (F.H.); gunda.ruzaike@uniklinik-freiburg.de (G.R.); ekkehart.lausch@uniklinik-freiburg.de (E.L.); anke.schumann@uniklinik-freiburg.de (A.S.); natascha.vdwerf-grohmann@uniklinik-freiburg.de (N.v.d.W.-G.); ute.spiekerkoetter@uniklinik-freiburg.de (U.S.); miriam.schmidts@uniklinik-freiburg.de (M.S.)

**Keywords:** *IGF2*, Silver–Russel syndrome, hypomethylation, hypoglycemia, growth hormone

## Abstract

Hypomethylation of *H19* and *IGF2* can cause Silver–Russell syndrome (SRS), a clinically and genetically heterogeneous condition characterized by intrauterine growth restriction, poor postnatal growth, relative macrocephaly, craniofacial abnormalities, body asymmetry, hypoglycemia and feeding difficulties. Isolated hypomethylation of *IGF2* has been reported in single cases of SRS as well. Here, we report on a 19-month-old patient who presented with two episodes of hypoglycemic seizures. No intrauterine growth restriction was observed, the patient did not present with SRS-typical facial features, and postnatal growth in the first months of life was along the lower normal percentiles. Exome sequencing did not reveal any likely pathogenic variants explaining the phenotype; however, hypomethylation studies revealed isolated hypomethylation of *IGF2*, while the methylation of *H19* appeared normal. Hypoglycemia responded well to growth hormone therapy, and the boy showed good catch-up growth. Our case demonstrates that SRS and isolated *IGF2* hypomethylation should be considered early in the diagnosis of recurrent hypoglycemia in childhood, especially in combination with small gestational age and poor growth.

## 1. Introduction

Hypomethylation of the imprinted telomeric control region 1 (ICR1) at 11p15.5 on the paternal allele is the most common genetically confirmed cause of Silver–Russel syndrome (SRS, OIMIM #180860), accounting for 35–50% of all cases [1,2,3,4,5]. Two genes lie within this region: *H19* and *IGF2* (insulin-like growth factor 2). Their expression is controlled by an intergenic differentially methylated region, DMR1, where an insulator between *H19* and *IGF2* inhibits *H19* enhancer effects on *IGF2* so that *IGF2* is not expressed. ICR1 methylation occurring only on the paternal allele abrogates the insulator effects so that the *H19* enhancer is able to activate *IGF2* expression. ICR1 hypomethylation on the paternal allele can therefore lower *IGF2* expression, resulting in the clinical picture of SRS [6,7]. For SRS cases, methylation studies often reveal hypomethylation of *H19* and *IGF2*, while only very few cases presenting with phenotypic characteristics overlapping SRS with isolated *H19* or *IGF2* hypomethylation have been reported to date [7]. Likewise, pathogenic paternal *IGF2* variants have been identified in SRS patients [8]. Rarely, SRS is caused by maternal uniparental disomy of chromosome 7 (mUPD7; 7–10% of SRS cases [9,10,11]), duplications, deletions, or translocations involving chromosome 7 and by pathogenic variants in *CDKN1C* (maternal allele) [12], *PLAG1* [13], and *HMGA2* [14]. Interestingly, about 40% of individuals who meet clinical criteria for SRS have negative molecular and/or cytogenetic testing [6].

SRS is characterized by severe asymmetric intrauterine growth retardation (IUGR), resulting in affected individuals being born small for gestational age (SGA) with relative macrocephaly at birth. Additional clinical features include craniofacial abnormalities such as a prominent forehead usually with frontal bossing and a triangular facies, fifth-finger clinodactyly, micrognathia with narrow chin, body asymmetry, and a variety of minor malformations [6]. Postnatal growth is also impaired, and the average adult height in untreated individuals is ~3.1 ± 1.4 SD below the mean [6]. Some patients with this condition show developmental delay (both motor and cognitive) and learning disabilities [6,15]. Children are at risk for fasting hypoglycemia, which can be explained by little subcutaneous fat, reduced body mass, feeding difficulties and poor appetite, and—in several children—by growth hormone (GH) deficiency [16].

Several clinical scoring systems for the diagnosis of SRS have been suggested [3,7,17,18], with the majority of them requiring the presence of three or four out of the following characteristic features: birth weight ≤-2SD, postnatal growth restriction, relative macrocephaly, facial characteristics and body asymmetry. The Netchine–Harbison Clinical Scoring System (NH-CSS), published in 2015, adds feeding difficulties as a sixth classical feature and requires four out of six criteria for the diagnosis of “likely SRS” [17].

Here, we report on a 19-month-old patient who presented with recurrent severe and symptomatic hypoglycemia. While he did not meet SRS clinical diagnostics criteria, we detected the isolated hypomethylation of *IGF2*. Although GH deficiency could not be demonstrated from peripheral blood samples, blood glucose concentrations stabilized under growth hormone treatment.

## 2. Methods

### 2.1. Genetic Diagnostics

Written consent was obtained from the patient’s parents.

### 2.2. Exome Sequencing

Exome sequencing and data analysis were performed as previously described [19,20]. In brief, DNA was extracted from 5 mL of ethylenediamine tetraacetic acid (EDTA) blood using a commercial kit (DNAeasy blood and tissue kit, Quiagen, Germantown, MD20874, USA). Two micrograms of DNA was used for library preparation and exome sequencing performed using Agilent sure select V6 kit. Variant filtering was performed for variants with a minor allele frequency below 1% in public databases such as gnomAD, 1000 genomes and ExAc as well as for coding variants and variants within 8 bp distance of splice sites.

### 2.3. Methylation-Specific Multiplex Ligation-Dependent Probe Amplification (MS-MLPA)

To investigate methylation within the IC2 (KvDMR) and IC1 (H19DMR) domains in the 11p15 BWS/RSS region, MS-MLPA was performed using SALSAMLPA probemix ME030 BWS; Russel Silver Syndrome (RSS) according the manufacturer’s protocol (MRC Holland, Amsterdam, The Netherlands). The kit contains 26 probes specific for the BWS/RSS 11p15 region with ten of these probes containing an HhaI recognition site and providing information on the methylation status of the BWS/RSS 11p15 region.

## 3. Case Presentation

The patient is the second child of non-consanguineous German parents. The four-year-old sister is healthy, but of short stature (<3rd centile at age 4 years). The mother suffered from gestational diabetes during both pregnancies, but otherwise no metabolic or endocrinological disorders were reported within the family. The patient was born at 40 + 2 weeks of gestation with a birth weight of 3100 g (11th centile) and a body length of 50 cm (12th centile). The boy was first presented to our metabolic clinic at the age of 19 months with a history of two hypoglycemic seizures within 2 weeks. Both events occurred after he had slept through the night for the first times. When the mother tried to wake him up in the morning, he was apathic and subsequently showed convulsions of his upper body. The parents gave him apple puree which led to rapid cessation of the seizure. While waiting for the emergency doctor, he ate yoghurt and sausage. Upon the arrival of the emergency team, all clinical symptoms had subsided, and the blood sugar was 4.4 mmol/L. A second very similar episode occurred 2 weeks later. The parents further reported that the boy used to wake up in the night demanding food.

Until the age of 19 months, he was normally developed, and no hypoglycemia-suggestive symptoms were reported. His growth was always within the lower normal range (see Figure 1).

His appetite was poor, but no special eating habits were reported. No febrile or gastrointestinal infections had occurred in the past. The local pediatrician recommended measuring the blood glucose concentration before breakfast for one week. This revealed two severe asymptomatic hypoglycemias of 2.1 mmol/L, respectively.

At presentation in the outpatient clinic, his clinical condition was very good. His body weight was 10.5 kg (15th centile) and his body length was 80 cm (6th centile) (Figure 1). The clinical examination was unremarkable. Laboratory work-up including metabolic investigations (urinary organic acids, acylcarnitines in dried blood spots, amino acids in serum) yielded normal results. Serum triglycerides (126 mg/dL, normal < 150 mg/dL), total cholesterol (134 mg/dL, normal 50–200 mg/dL) and lactic acid (1.5 mmol/L, normal < 1.6 mmol/L) were normal. Serum IGF1 and IGFBP3 levels were also within the reference range. GH deficiency was excluded by an arginine stimulation test (30 min after arginine supplementation growth hormone concentration 11.3 ng/mL; normal > 8.0 ng/mL). An oral glucose tolerance test was performed and no signs of impaired glucose tolerance or insulin resistance were observed. Abdominal sonography was also normal. Further diagnostic work-up including whole exome sequencing and methylation studies for SRS was initiated. Until the results became available, symptomatic treatment with uncooked cornstarch at bedtime and during the night was started and no further symptomatic hypoglycemic events were observed; however, continuous glucose monitoring with a Freestyle libre^®^ 2 device (subcutaneous glucose sensor by Abbot) was implemented and confirmed recurrent, severe but asymptomatic hypoglycemia.

Whole exome sequencing yielded no likely pathogenic variants explaining the phenotype. Methylation studies (MLPA) of the region 11p15, the critical methylation-sensitive region in patients with SRS, yielded normal results for four H19DMR probes, but the signal of the *IGF2* probe was decreased to 50% compared to healthy controls, suggesting hypomethylation of the paternally expressed *IGF2* gene (Figure 2).

GH therapy was therefore initiated at the age of 2 years and led to significant stabilization of glucose homeostasis. Cornstarch supplementation could be stopped, and no further severe hypoglycemias have occurred so far. The boy showed a good catch-up growth of 4 cm within the first 3 months of GH treatment.

## 4. Discussion

We report on a patient with *IGF2* hypomethylation who presented with severe recurrent hypoglycemia at the age of 19 months. The 11p15 region comprises two imprinted domains that are important for the control of fetal and postnatal growth, ICR1 and ICR2 [7]. ICR1 coordinates the expression of two oppositely imprinted genes, *H19* and *IGF2*. A study by Bartholdi et al. demonstrated that the majority of SRS patients with methylation abnormalities show hypomethylation at both the *H19* and *IGF2* genes [7]. Only a small subset of patients carries epimutations restricted to either the *H19* or *IGF2* gene as seen in our patient, in whom only hypomethylation of *IGF2* was detected [7]. Due to the low number of reported cases, it is not clear if the phenotypes in cases with isolated *IGF2* hypomethylation and cases with hypomethylation of the entire region are comparable. Importantly, our patient did not meet SRS clinical diagnosis criteria (no IUGR/small for gestational age, low–normal postnatal growth, no typical craniofacial features, and no body asymmetries). We performed MLPA analyses to rule out SRS as a cause of the severe hypoglycemic episodes.

It is well known that children with SRS are prone to hypoglycemia, especially after prolonged fasting [6,21]. The precise mechanism of hypoglycemia in SRS and likewise *IGF2* dysfunction is not clear to date, but several factors are discussed: *IGF2* presumably is able to bind to both IGF1 and *IGF2* receptors as well as the insulin receptor and interacts with IGFBP3, possibly influencing glucose homeostasis [16]. It is further assumed that lower GH levels in SRS may contribute to a reduced fasting tolerance in SRS. Young children with SRS have little subcutaneous fat, low muscle and liver mass, and a disproportionately large brain-for-body size [16,21]. Additionally, many patients have poor appetite, oral motor problems, and feeding disorders [22]. GH deficiency has been described in several cases [16]. The incidence of hypoglycemia is approximately 27% [15], with a high frequency of spontaneous, asymptomatic nocturnal hypoglycemia [16,21]. Although hypoglycemia is quite common in SRS patients, many remain asymptomatic; therefore, hypoglycemic seizures as observed in our patient are usually not the first presenting symptom. It can be assumed that our patient had recurrent asymptomatic hypoglycemia before he finally presented with a hypoglycemic seizure. In several studies, good response to GH treatment in SRS children with respect to growth was shown [23,24]. The goals of GH treatment, however, are not only to improve growth velocity but also to improve body composition, psychomotor development, and appetite, to optimize linear growth, to increase lean body mass and muscle power and to reduce the risk of hypoglycemia [21]. In our patient, GH treatment was mainly started to better control hypoglycemia. Indeed, GH therapy resulted in a marked improvement of glucose homeostasis.

## 5. Conclusions

Our case demonstrates that severe hypoglycemia may be the first presenting symptom for SRS-like cases associated with *IGF2* hypomethylation. SRS and *IGF2* hypomethylation should be considered early in the differential diagnosis of recurrent ketotic hypoglycemia in childhood, as patients benefit from early GH treatment.

## Figures and Tables

**Figure 1 diagnostics-11-00749-f001:**
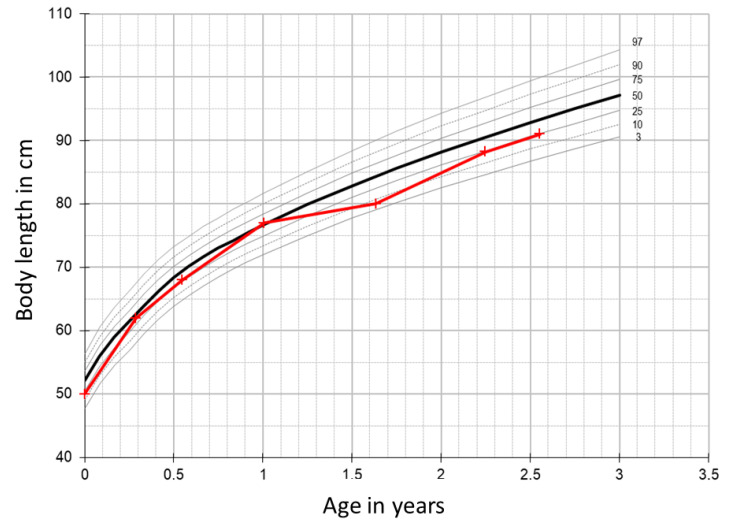
Growth chart of the patient. The boy was not small for gestational age, and his further growth was within the lower normal range.

**Figure 2 diagnostics-11-00749-f002:**
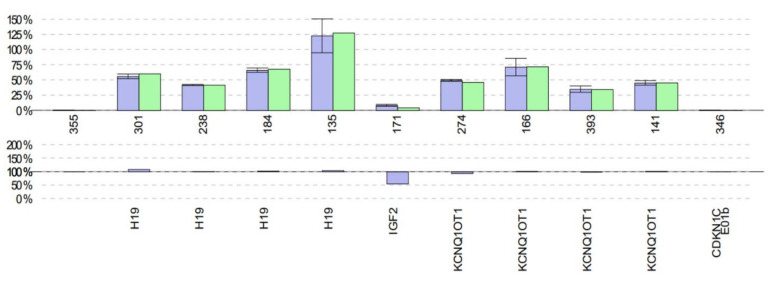
*IGF2* hypomethylation revealed by H19DMR MLPA. While no difference between patient (green) and controls (purple) (upper panel) was observed for four *H19*, four *KCNQ1OT1* and one *CDKN1C* probe, *IGF2* methylation was 50% lower in the patient compared to the controls (lower panel, purple).

## Data Availability

Not applicable.

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
