# Peer review of "Isolated Hypomethylation of IGF2 Associated with Severe Hypoglycemia Responsive to Growth Hormone Treatment"

_diagnostics, 2021, doi:10.3390/diagnostics11050749_

Round 1

Reviewer 1 Report

The case described by Grünert et al. is a fascinating clinical case, as SRS and SRS-like syndrome, which is extremely heterogeneous and molecular aetiology is not clear a high percentage of patients. Also, identify molecular mechanisms of severe hypoglycemia associated with SRS -Like in children, it could benefit from early GH treatment.

1.-The authors have considered measurement serum IGF-2 previous and after GH treatment?.

1.-Current terminology should be used and not mixed with old terminology as:

ICR1  (old) and CDKN1C (new).

2.-The Netchine-Harbison Clinical Scoring System (NH-CSS), was published in 2015 not 2016

3.- There are some word-paragraph wrong as:

17, 62:   IFG2 for IGF2

65-79:. This paragraph have no sense

  1. Materials and Methods. The Materials and Methods should be described with sufficient details to allow oth- 65 ers to replicate and build on the published res etc.

Author Response

Reviewer 1

The case described by Grünert et al. is a fascinating clinical case, as SRS and SRS-like syndrome, which is extremely heterogeneous and molecular aetiology is not clear a high percentage of patients. Also, identify molecular mechanisms of severe hypoglycemia associated with SRS -Like in children, it could benefit from early GH treatment.

 We thank the reviewer for his positive assessment.

1.-The authors have considered measurement serum IGF-2 previous and after GH treatment?

Unfortunately, serum IGF 2 has never been measured in our patient as it is not available as a routine marker.

1.-Current terminology should be used and not mixed with old terminology as:

ICR1  (old) and CDKN1C (new).

We thank the reviewer for this question. However we feel this is a misunderstanding: ICR1 describes a genomic region to which the H19 and IGF2 locate. CDKN1C is a gene located in the ICR2 genomic region.

2.-The Netchine-Harbison Clinical Scoring System (NH-CSS), was published in 2015 not 2016

            The reviewer is perfectly right. We have changed this in the introduction.

3.- There are some word-paragraph wrong as:

17, 62:   IFG2 for IGF2

            This typos have been corrected.

65-79:. This paragraph have no sense

  1. Materials and Methods. The Materials and Methods should be described with sufficient details to allow oth- 65 ers to replicate and build on the published res etc.

This paragraph was inserted by mistake by the editorial office during transfer of our submitted Word document in the journal’s template. We have deleted this paragraph now.

Reviewer 2 Report

 This is an interesting study that provides information about the case reporting a Isolated hypomethylation of IGF2 associated with severe hypo-2 glycemia responsive to growth hormone treatment.

 There are some topics in the article that should be considered

Introduction is too long, there is no mentioned the aim of the study

By mistake after the line 59 the authors have a part of the Journal author`s instructions  it must be corrected

The lines 59-64 I consider must be moved to methodology

Line 85 the mentioned “Written consent was obtained from the patient`s parents” is not regarding Genetic diagnostics but it concern the ethics of the study. There is some permission from the ethic committee?

It could be It could be useful to present a complete biochemical picture of the patient including lipidemic profile and galactic acid levels

In the discussion, the mechanism by which the hypomethylation of IGF-1 could produce hypoglicema should be analyzed.

Plagiarisms was detected but there is only 3% plagiarism in the introduccion and 5% plagiarism in the discussion

Author Response

Reviewer 2

This is an interesting study that provides information about the case reporting a Isolated hypomethylation of IGF2 associated with severe hypoglycemia responsive to growth hormone treatment.

We thank reviewer 2 for his positive evaluation of our case report.

There are some topics in the article that should be considered

Introduction is too long, there is no mentioned the aim of the study

We think that the information provided in the introduction is necessary for the reader to understand the unusual findings in our patient. As the last (14 lines) paragraph of the introduction has been inserted by the editorial office by mistake when transfering the manuscript to the journal’s template, the introduction is now much shorter.

By mistake after the line 59 the authors have a part of the Journal author`s instructions  it must be corrected

            This has been corrected.

The lines 59-64 I consider must be moved to methodology

This text has been deleted as it was not part of our original  manuscript, but has been inserted by mistake (see above).

Line 85 the mentioned “Written consent was obtained from the patient`s parents” is not regarding Genetic diagnostics but it concern the ethics of the study. There is some permission from the ethic committee?

As this is a single case report and no research study, no ethic approval is needed in Germany. Nevertheless, we asked the parents to give their written consent for the publication of this case report. This  is also mentioned in the Declarations section at the end of the article:

Institutional Review Board Statement: Ethics approval and consent to participate: Not applicable

Informed Consent Statement: The patients’ parents gave their informed consent for publication.

It could be It could be useful to present a complete biochemical picture of the patient including lipidemic profile and galactic acid levels

We have added the following information to the case report section: “Serum triglycerides (126 mg/dL, normal < 150 mg/dL), total cholesterol (134 mg/dL, normal 50-200 mg/dL) and lactic acis (1.5 mmol/L, normal < 1.6 mmol/l) were normal.”

In the discussion, the mechanism by which the hypomethylation of IGF-1 could produce hypoglicema should be analyzed.

We thank the reviewer for raising this topic and have adapted the paragraph on hypoglycemia as follows:

It is well known that children with SRS are prone to hypoglycemia, especially after prolonged fasting [6, 21]. The precise mechanism of hypoglycemia in SRS and likewise IGF2 dysfunction is not clear to date, but several factors are discussed: IGF2 presumably is able to bind to both IGF1 and IGF2 receptors as well as the insulin receptor and interacts with IGFBP3, possibly influencing glucose homeostasis [16]. It is further assumed that lower GH levels in SRS may contribute to a reduced fasting tolerance in SRS. Young children with SRS have little subcutaneous fat, low muscle and liver mass, and a disproportionately large brain-for-body size [16, 21]. Additionally, many patients have poor appetite, oral motor problems, and feeding disorders [22]. GH deficiency has been described in several cases [16].

Plagiarisms was detected but there is only 3% plagiarism in the introduccion and 5% plagiarism in the discussion

We have rechecked again that all citations are correctly cited.
